# WeInfer: Unleashing the Power of WebGPU on LLM Inference in Web Browsers

Submission Id: 1041

## Abstract

Web-based large language model (LLM) has garnered significant attention from both academia and industry due to its potential to combine the benefits of on-device computation with the accessibility and portability of Web applications. The advent of WebGPU, a modern browser API that enables Web applications to access and utilize a device's GPU, has opened up new possibilities for GPU-accelerated LLM inference within browsers. Several frameworks have been developed to support Web-based LLM inference with WebGPU. However, our experiment reveals that these frameworks exhibit inefficiencies in GPU utilization, influencing the LLM inference speed. These inefficiencies primarily arise from underutilizing the full capabilities of WebGPU, particularly in resource management and execution synchronization. To address these limitations, we present WeInfer, an efficient Web-based LLM inference framework specifically designed to unleash the power of WebGPU. WeInfer incorporates two key innovations: 1) buffer reuse strategies that reduce the overhead associated with resource preparation, optimizing the lifecycle management of WebGPU buffers, and 2) an asynchronous pipeline that decouples resource preparation from GPU execution, enabling parallelized computation and deferred result fetching to improve overall efficiency. We conduct extensive evaluations across 9 different LLMs and 5 heterogeneous devices, covering a broad spectrum of model architectures and hardware configurations. The experimental results demonstrate that WeInfer delivers substantial improvements in decoding speed, achieving up to a 3.76× performance boost compared with WebLLM, the state-of-the-art Web-based LLM inference framework.

## CCS Concepts

• **Information systems** → **Web applications**; • **Computer systems organization** → **Neural networks**.

## Keywords

Large Language Model, WebGPU, Inference Acceleration

## 1 Introduction

Large Language Models (LLMs) [7, 11, 34] have revolutionized both academia and industry, with models like ChatGPT [47] demonstrating unprecedented capabilities in natural language generation. As the scope of LLM applications continues to expand [10, 26], the demand for reliable, efficient, and scalable deployment has become increasingly urgent. One promising direction is to perform LLM inference directly within Web browsers, commonly referred to as Web-based LLM inference [30].

Web-based LLM inference effectively addresses several limitations inherent to cloud-based deployment [3, 36, 46], particularly by mitigating privacy concerns [37, 42, 43] and reducing network latency [9]. Meanwhile, compared with deploying LLM directly on the client's native OS, Web-based LLMs benefit from the cross-platform compatibility of modern Web browsers [35, 54], enabling scalable and decentralized deployment across heterogeneous edge devices, offering wide accessibility and portability.

Several inference frameworks, pioneered by MLC-AI, Google and Hugging Face [12, 15, 30], have been developed to facilitate Web-based LLM inference. These frameworks such as WebLLM and MediaPipe LLM enable GPU acceleration through leveraging WebGPU [44], a modern API that provides Web applications with access to GPU hardware [6]. This approach represents a significant advancement over JavaScript-based execution by harnessing the parallel processing power of GPUs.

Despite these advancements, our measurement study reveals that existing Web-based LLM inference frameworks exhibit suboptimal GPU utilization, limiting the decoding speed of LLM inference. We identify inefficient use of WebGPU as a major contributing factor to this performance degradation. Specifically, existing frameworks spend excessive time on preparing WebGPU resources. Additionally, these frameworks rely on synchronous execution models that fail to take advantage of WebGPU's advanced features, which results in unnecessary blocking and further performance loss.

To overcome these limitations, we propose **We**bGPU-centric **Infer**ence (**WeInfer**), a novel Web-based LLM inference framework that introduces two key optimization strategies specifically aimed at reducing inefficiencies associated with WebGPU. Our approach focuses on: 1) minimizing the overhead related to resource preparation before GPU execution, and 2) reducing synchronous blocking and the latency incurred when fetching results from the GPU.

For the first optimization, WeInfer introduces buffer reuse strategies that optimize WebGPU's buffer lifecycle management, significantly reducing the overhead associated with preparing WebGPU resources. The second optimization is more challenging due to the auto-regressive nature of LLMs, where each prediction relies on the token generated in the previous step, complicating the introduction of asynchronous computation. WeInfer tackles this issue by leveraging WebGPU's timeline model to decouple resource preparation from GPU execution while postponing result fetching. This approach allows WeInfer to implement a parallel asynchronous pipeline, effectively eliminating unnecessary blocking during inference and improving overall efficiency.

Unlike prior approaches that focus on accelerating Web-based LLMs through operator- or model-level optimizations, such as automatic operator tuning [8], subgraph fusion [29], and subgroup cooperation [17], WeInfer specifically targets inefficiencies related to WebGPU. This WebGPU-centric approach addresses the unique constraints imposed by Web browsers, where WebGPU operates through a single, shared GPU process [18, 20]. WeInfer introduces optimizations solely at the API level, without altering the underlying model architectures, operators, or inference algorithms. This

modular design allows WEINFER to integrate seamlessly with existing acceleration techniques.

We conduct comprehensive evaluations across 9 LLMs and 5 heterogeneous devices. These models span parameter sizes ranging from 135 million to 7 billion, representing 3 mainstream architectures. The selected devices include diverse GPUs with different operation systems, encompassing a broad range of computational capabilities. The experimental results demonstrate that compared with WebLLM, WEINFER achieves a minimum 1.34× improvement in decoding speed on mid-range GPUs for models with fewer than 1.5 billion parameters, improving decoding speed from 24.18 ms/token to 18.07 ms/token, with performance gains reaching up to 3.76× on high-end GPUs like the RTX 4090, achieving decoding speed of 8.43 ms/token. Further ablation studies confirm that each of our proposed optimization techniques contributes significantly to the observed performance improvements.

In summary, this paper makes the following key contributions[1]:

- To our best known, we first identify the unique inefficiencies in existing Web-based LLM inference frameworks, particularly in the decoding stage, where GPU utilization is constrained by suboptimal use of WebGPU.
- We propose WEINFER, a Web-based LLM inference framework with novel optimizations that leverages advanced features of WebGPU to reduce resource preparation overhead and eliminate blocking, significantly accelerating the decoding stage.
- We conduct extensive evaluations across diverse LLMs and hardware, demonstrating significant performance improvements in heterogeneous environments. Additionally, ablation studies and hyper-parameter sensitivity analysis validate the effectiveness of each optimization and the robustness of WEINFER across various scenarios.

## 2 Background

In this section, we present the background knowledge of LLM inference and outline the workflow of inferring LLMs via WebGPU in Web browsers.

### 2.1 Inference of Large Language Model

LLMs commonly adopt a multi-layer architecture containing Transformer layers, each comprising various operators parameterized by model weights. A LLM with $N$ layers is represented as a nested function: $\text{LLM}_{\Theta}() = f^1_{\theta_1}\left(f^2_{\theta_2}\left(\ldots f^N_{\theta_N}\ldots\right)\right)$, where $\Theta = \{\theta_1, \theta_2, \ldots, \theta_N\}$ represents the model parameters, while $f^i_{\theta_i}$ depicts the $i$-th layer with corresponding parameters $\theta_i$.

Upon receiving an input sequence of tokens $\mathbf{x} = [x_1, x_2, \ldots, x_L]$ with a length of $L$, the LLM processes the input and predicts the initial token $y_0 = \text{LLM}_{\Theta}(\mathbf{x})$, referred to as the prefill stage. The predicted token $y_0$ then serves as input for the decoding stage, where predictions are generated in an auto-regressive manner as $y_i = \text{LLM}_{\Theta}(y_{i-1})$.

The data dependency between predictions across multiple decoding steps can be expressed as:

$$y_i = \text{LLM}_{\Theta}(y_{i-1}) \xrightarrow{\text{finish}} y_{i+1} = \text{LLM}_{\Theta}(y_i), \quad (1)$$

---

[1]WEINFER will be open-source once this paper is accepted.

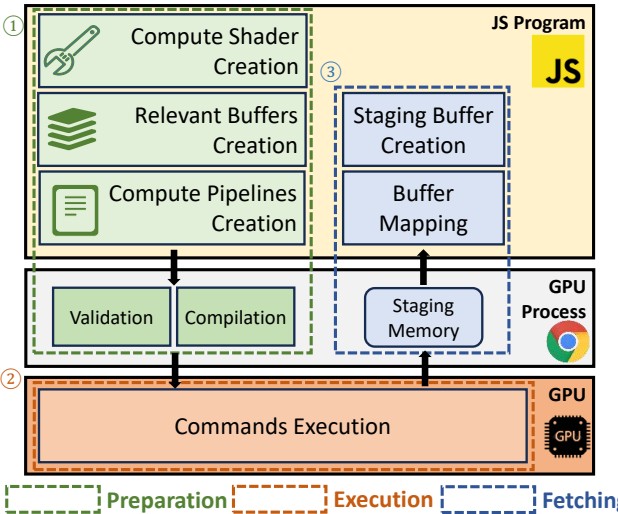

**Figure 1: Workflow of conducting computational tasks for LLM inference using WebGPU**

where the symbol A $\xrightarrow{\text{finish}}$ B represents that operation A enables operation B to commence. Similarly, the dependencies for each predicted token throughout multiple decoding steps can be represented as:

$$z^1_i = f^1_{\theta_1}\left(t^1_i\right) \xrightarrow{\text{finish}} z^2_i = f^2_{\theta_2}\left(z^1_i\right) \rightarrow \ldots \rightarrow z^N_i = f^N_{\theta_N}\left(z^{N-1}_i\right), \quad (2)$$

where $z^j_i$ signifies the output of the $j$-th neural network layer during the prediction of the $i$-th token, culminating in the predicted token ID $y_{i+1} = z^N_i = f^N_{\theta_N}\left(z^{N-1}_i\right)$.

### 2.2 Workflow of Inferring LLM Using WebGPU

Web-based LLM inference frameworks, implemented as JavaScript programs, enable LLM inference directly within browsers, leveraging GPU acceleration via WebGPU. The workflow for performing computational tasks in LLM inference using WebGPU is illustrated in Figure 1 and consists of three main stages:

The inference framework collaborates with the GPU process to set up crucial resources. Necessary resources include compute shaders which define LLM operators, data buffers storing weights and input, and uniform buffers for shader interpretation metadata. Validation and compilation of these resources are managed by the GPU process, a kernel process shared by all Web applications.

- **Preparation** (Figure 1 ①): The inference framework collaborates with the GPU process to set up crucial resources such as compute shaders defining LLM operators, data buffers storing weights and input, and uniform buffers for shader interpretation metadata. These resources undergo validation and compilation by the GPU process, a kernel process shared by all Web applications.
- **Execution** (Figure 1 ②): Computation begins on the GPU once all necessary resources are in place.

**Table 1: Time cost (ms) of different stages and GPU utilization (abbreviated as GPU Util.) during a decoding step of existing Web-based LLM inference frameworks**

| Framework | $T$ | $T_{\text{Prep}}^{\text{C,D}}$ | $T_{\text{Exec}}^{\text{Q}}$ | $T_{\text{Fetch}}^{\text{C,D}}$ | GPU Util. |
|---|---|---|---|---|---|
| **MediaPipe LLM** | 33.16 | 5.41 | 10.17 | 20.53 | 30.67% |
| **WebLLM** | 61.83 | 11.01 | 43.33 | 4.45 | 70.08% |

- **Fetching** (Figure 1 ③): Upon the completion of computation, the fetching stage retrieves results by mapping GPU memory to the CPU through a staging buffer, facilitating the transfer of output data back to the application.

These stages operate across different processes and hardware. WebGPU utilizes a timeline model [45] to delineate operations flow under the single GPU process architecture of the browser, which involve three distinct timelines:

- **Content timeline**, executing Web application code.
- **Device timeline**, handling resources processed by the GPU process.
- **Queue timeline**, representing computation on GPU hardware.

The preparation and fetching align with the content and device timelines, whereas the execution stage is confined to the queue timeline. The time overhead for each stage per decoding step is represented by $T_{\text{Prep}}^{\text{C,D}}$, $T_{\text{Exec}}^{\text{Q}}$, and $T_{\text{Fetch}}^{\text{C,D}}$, where the superscripts indicate the associated timelines (C: content timeline, D: device timeline, Q: queue timeline). As execution depends on the associated resources to start, we denote the time delay due to waiting for preparation as $\delta T_{\text{Prep}}^{\text{C,D}}$, where $\delta > 0$ indicates extended validation and data transfer leading to delays. These stages are sequenced by Web-based LLM inference frameworks as shown in Figure 3(a). In this sequential workflow, the time spent per decoding step can be approximated as

$$T = \delta T_{\text{prep}}^{\text{C, D}} + T_{\text{exec}}^{\text{Q}} + T_{\text{fetch}}^{\text{C, D}}. \tag{3}$$

## 3 Measurement Study

We delve into a measurement study to investigate how the characteristics of the Web environment impact the inference performance of Web-based LLMs.

**Setup**. Our experiments center on two widely-used Web-based LLM inference frameworks: MediaPipe LLM [15] and WebLLM [30]. We exclude Transformer.js as it has not officially introduced WebGPU support. We perform inference on their demonstration pages [16, 31] using Chrome on a Windows system equipped with a GTX 1660 Ti GPU. Specifically, WebLLM employs the Qwen2-0.5B-q4f16 model, while MediaPipe LLM utilizes the Gemma 2B model. By tracing the inference processes via Chrome DevTools, we gauge the decoding cost $T$ and the time overhead for key stages of inference, namely $T_{\text{Prep}}^{\text{C,D}}$, $T_{\text{Exec}}^{\text{Q}}$ and $T_{\text{Fetch}}^{\text{C,D}}$ to assess the efficiency of WebGPU utilization in these frameworks. Table 1 presents our results.

**Result Analysis**. Our observations demonstrate shortcomings in the overall GPU efficiencies of existing inference frameworks. MediaPipe LLM exhibits a mere 30% GPU utilization, whereas WebLLM showcases a relatively superior 70% GPU utilization. Notably, MediaPipe LLM spends 20.53ms on synchronous fetching during a decoding step, significantly impairing GPU efficiency. In contrast, WebLLM curtails the fetching overhead to around 4.45ms but still encounters synchronous blocks. Furthermore, MediaPipe LLM and WebLLM spend 5.41ms and 11.01ms, respectively, on preparation for each prediction, delaying the initiation of GPU computation, which depends on these resources. We also identify that conducting post-processing (i.e. token biasing and sampling) on CPU which is adopted by WebLLM further impedes GPU efficiency and inflates the total time $T$, an issue addressed by MediaPipe LLM by shifting post-processing to the GPU.

Through analyzing performance traces from Chrome DevTools, we identify that inefficiencies primarily occur during the preparation and fetching stages. These inefficiencies arise due to the suboptimal computational pattern that follows a sequential workflow when leveraging WebGPU. The preparatory burden, involving WebGPU validation and compilation, leads to increased $\delta T_{\text{Prep}}^{\text{C,D}}$. Similarly, frequent blocking during the fetching stage further reduces GPU efficiency.

To unleash the full potential of WebGPU, we propose optimization strategies targeting the acceleration of the costly preparation and fetching stages. Our strategy revolves around 1) expediting the start of the execution stage by reducing the amount of resources to be prepared and 2) mitigating the fetching stage overhead by postponing fetching while asynchronously parallelizing the preparation with execution to harness different timelines concurrently.

## 4 WeInfer

We first introduce the overview of WeInfer (Section 4.1). Then we detail our optimizations addressing the unique challenges of the Web environment (Section 4.2, 4.3). Further details of the browser constraints on leveraging GPU are presented in Appendix A.

### 4.1 Overview

WeInfer is designed to enhance the workflow of existing Web-based LLM inference frameworks by introducing two key optimizations: 1) WebGPU uniform buffer reuse, and 2) an asynchronous pipeline to parallelize preparation and execution. Figure 2 provides an overview of WeInfer with these optimizations. Algorithm 1 demonstrates the workflow of WeInfer.

Initially, WeInfer follows the same procedure as the base framework, loading the LLM and user input and conducting the prefill stage. Once WeInfer enters the decoding stage, a scheduler restructures the preparation and fetching processes. First, the task issuer is scheduled to continuously prepare resources and issue computational tasks (①), leveraging a resource cache to avoid redundant creation of static buffers that remain constant throughout inference (②). WeInfer eliminates blocking by letting task issuer continuously work on preparation without waiting for GPU. While the CPU handles preparation, the GPU executes computation in parallel (③). Once sufficient predictions have accumulated, WeInfer schedules the fetcher to retrieve results through buffer mapping (④), reducing the number of costly fetch operations.

### 4.2 WebGPU Uniform Buffer Reuse

To accommodate the sandbox mechanism of browsers [18, 19], WebGPU enforces strict validation within the GPU process when

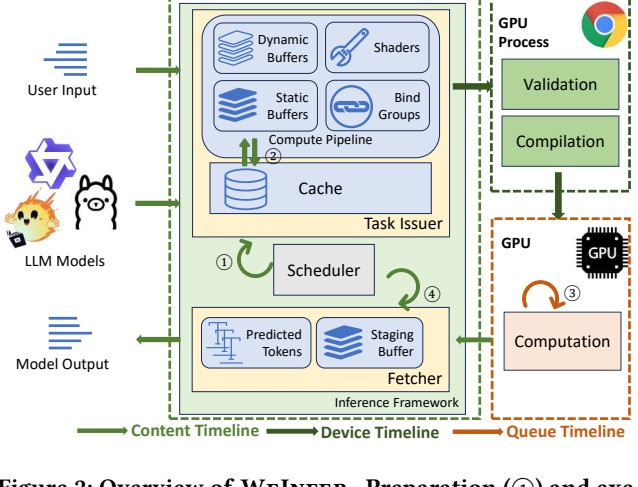

**Figure 2: Overview of WeInfer . Preparation (①) and execution (③) are scheduled in parallel, while fetching (④) is postponed until multiple predictions are in place. Caching is introduced to accelerate preparation process (②)**

**Table 2: Required uniform values of a Transformer layer in the decoding stage**

| Operator | Required Uniform Values |
| --- | --- |
| **Layer Normalization** | $1, embedDim$ |
| **Self-Attention** | |
| Projection of Q, K, V and Output | $1, embedDim$ |
| Attention | $numHead, seqLen, headDim$ |
| **Feed Forward Network** | $1, embedDim, interDim$ |
| **Residual** | $1, embedDim$ |

preparing resources including data buffers [20]. Therefore, we try to reduce the amount of buffers that are needed to be prepared to reduce the preparation overhead $\delta T_{prep}^{C, D}$. To assess how WebGPU buffer creations impact the preparation and devise reuse strategies, we analyze the lifecycles of buffers involved in the inference and categorize buffers into four distinct types:

- **Weight Buffers**: These buffers store the LLM's parameters and are integral to the execution of the compute shaders. Weight buffers are loaded and transferred to the GPU once when the model initiates, and remain constant during the inference. Therefore, their creation does not impact the preparation time in the decoding steps.
- **Temporary Buffers**: These buffers hold intermediate results for the compute shaders with fixed sizes across decoding steps. Therefore, temporary buffers can be created once during the prefill stage and reused throughout the decoding process, impacting less to the preparation cost.
- **Staging Buffers**: These buffers are required to fetch results (such as logits or token IDs) from the GPU to the CPU. Staging buffers do not influence the preparation cost since they are created after the execution stage, allowing their associated costs to be covered.

**Algorithm 1** WeInfer: Accelerated LLM inference utilizing WebGPU features

**Input:** User input text $input$, weights $W = \{w_1, \ldots, w_N\}$ and shader codes $S = \{s_1, \ldots s_N\}$ of operators in the selected LLM
**Output:** Predicted output text $output = \{y_0, y_1, y_2, \ldots, y_M\}$
1: $weightBuffers \leftarrow$ createWeightBuffers($W$)
2: $operatorShaders \leftarrow$ compileShaderModules($S$)
3: $y_0, inputBuffer, resultBuffer, tempBuffers \leftarrow$
$\qquad\qquad\qquad$ prefill($input, weightBuffers$)
4: $resourceCache \leftarrow \emptyset$
5: $output \leftarrow \{y_0\}$
6: **for** $i = 1 \rightarrow M$ **do**
7: $\quad$ **for** $j = 1 \rightarrow N$ **do** $\qquad\qquad$ ▷ Task issuer is scheduled
8: $\qquad resources \leftarrow \{weightBuffers[j], tempBuffers[j]\}$
9: $\qquad$ **if** $j = 1$ **then**
10: $\qquad\quad resources$.addResources($inputBuffer$)
11: $\qquad$ **end if**
12: $\qquad$ **if** $j = N - 1$ **then**
13: $\qquad\quad resources$.addResources($resultBuffer$)
14: $\qquad$ **end if**
15: $\qquad uniformValues =$ getRequiredUniform($j$)
16: $\qquad$ **if** $i = 1$ **then** $\qquad$ ▷ Cache in the first decoding step
17: $\qquad\quad uniformBuffers \leftarrow$ createUniformBuffers($s_j$)
18: $\qquad\quad resourceCache$.update($j, uniformBuffers$)
19: $\qquad\quad resources$.add($uniformBuffers, uniformValues$)
20: $\qquad$ **else**
21: $\qquad\quad uniformBuffers \leftarrow$
$\qquad\qquad\qquad resourceCache$.get($uniformValues$)
22: $\qquad\quad resources$.addResources($uniformBuffers$)
23: $\qquad$ **end if**
24: $\qquad resources$.addResources($operatorShaders$.get($j$))
25: $\quad$ **end for**
26: $\quad computePass \leftarrow$ createWebGPUComputePass($resources$)
27: $\quad$ submitComputePass($computePass$) ▷ Eliminate blocking
28: $\quad$ **if** $i \% threshold = 0$ or $i = M$ **then** ▷ Postpone fetching
29: $\qquad predTokenIds \leftarrow$ waitForFetching($resultBuffer$)
30: $\qquad output$.append($predTokenIds$)
31: $\quad$ **end if**
32: **end for**

- **Uniform Buffers**: These buffers describe metadata for the compute shaders, such as the shapes of input/output data and other control parameters necessary for GPU execution. Each operator in an LLM requires its own uniform buffer, leading that the frequent creation of these buffers in every decoding step imposes significant overhead.

WeInfer implements a caching mechanism to mitigate the overhead caused by frequent uniform buffer creation. This mechanism leverages the fact that tensor shapes are consistent across different operators and decoding steps. As shown in Table 2, many operators, such as layer normalization and residual operators, require uniform buffers with the same values (e.g. $1, embedDim$), making recreating these uniform buffers for different operators redundant. Additionally, since $embedDim$ remains consistent throughout decoding, these buffers can be shared across operators and decoding iterations. We call these buffers static buffers and reuse them as

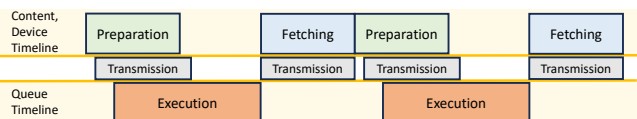

(a) Sequential workflow of inferring Web-based LLM

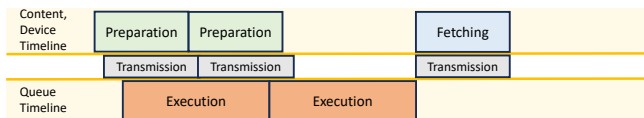

(b) Asynchronous pipeline that parallelizes preparation and executing

**Figure 3: Comparison of the sequential workflow and our asynchronous pipeline. Perform decoding synchronously limits the GPU utilization, while our approach increases efficiency by parallelizing operations across different timelines**

shown in Figure 2 ②, reducing the preparation cost $\delta T_{\text{prep}}^{\text{C, D}}$ by eliminating redundant buffer creation.

## 4.3 Asynchronous Pipeline for Parallelization of Preparation and Execution

Unlike native environments, modern browsers like Chrome utilize a process-based isolation architecture, restricting Web applications to access GPU through a single, shared GPU process [5, 13, 32]. The unique GPU process holds the GPU driver, resulting in that WebGPU requires an additional staging step for fetching results as GPU buffers are mapped only to the GPU process's memory space, necessitating further steps to transfer data to the application [20]. Therefore, to reduce the burden of frequently fetching GPU buffers, WeInfer implements an asynchronous pipeline (Figure 3(b)) where GPU and CPU continuously work in parallel to postpone fetching.

However, the inherent data dependencies (Equation (1)) in LLM inference pose challenges to parallelizing preparation and execution as each prediction depends on the result of the previous prediction. To address this challenge, we begin with analyzing the dependencies of WebGPU operations on different timelines. Through decoupling independent operations across different timelines by leveraging the features of WebGPU, WeInfer enables an asynchronous pipeline (Figure 3(b)) that mitigates blocking and enables GPU to keep on execution without retrieving the actual value of the last prediction.

The data dependency (2) during a decoding step dictates the order of tasks being computed for different layers, leading to the operation dependency occurring on the WebGPU queue timeline, represented as:

$$\text{compute}(r_j)^{\text{Q}} \xrightarrow{\text{finish}} \text{compute}(r_{j+1})^{\text{Q}}, \qquad (4)$$

where the superscript Q indicates operations occur on the queue timeline and $r_j$ denotes the computational resources needed for layer $f_{\theta_j}$. Meanwhile, the preparation of $r_j$ is required to be accomplished before execution, leading to the operation dependency expressed as:

$$\hat{r}_j = \text{issue}\left(f_{\theta_j}\right)^{\text{C}} \xrightarrow{\text{finish}} r_j = \text{process}\left(\hat{r}_j\right)^{\text{D}} \qquad (5)$$

$$r_j = \text{process}\left(\hat{r}_j\right)^{\text{D}} \xrightarrow{\text{finish}} \text{compute}(r_j)^{\text{Q}} \qquad (6)$$

where $\hat{r}_j$ represents original computational resources submitted by the inference framework which is expected to be processed by the GPU process. At the end of computation, there exists dependency that fetching is only available when all computations relevant to the target buffer get accomplished:

$$\left\{\text{compute}\left(r_j\right) \mid \forall \text{ issued } r_j\right\}^{\text{Q}} \xrightarrow{\text{finish}} y_{i+1} = \text{fetch}\left(z_i^N\right)^{\text{C, D}} \qquad (7)$$

Our key insight is that the logical data dependency throughout decoding steps (1) does not impose restrictions to either preparation on the content timeline and device timeline or execution on the queue timeline, as these operations are only confined by dependencies (4)(5)(6). Existing approaches overlook the asynchronism among different WebGPU timelines, invoking an `await` statement after each decoding step to synchronously wait for the retrieval of the token ID before proceeding to the next iteration. This unnecessarily pose operation dependencies represented as:

$$y_{i+1} = \text{fetch}\left(z_i^N\right)^{\text{C}} \xrightarrow{\text{finish}} \hat{r}_j = \text{issue}\left(f_{\theta_j}\right)^{\text{C, D}} \qquad (8)$$

$$\left\{\text{compute}\left(r_j\right) \mid \forall \text{ issued } r_j\right\}^{\text{Q}} \xrightarrow{\text{finish}} \hat{r}_{j+1} = \text{issue}\left(f_{\theta_{j+1}}\right)^{\text{C, D}} \qquad (9)$$

These dependencies (8)(9) occur across timelines, leading to blocking as CPU needs to wait for lengthy computations on the GPU. The overall operation dependencies can be represented as:

$$\text{LLM}_{\Theta}\left(y_i\right)^{\text{C}} \xrightarrow{\text{finish}} \text{LLM}_{\Theta}\left(y_i\right)^{\text{D}} \xrightarrow{\text{finish}} \text{LLM}_{\Theta}\left(y_i\right)^{\text{Q}} \qquad (10)$$

$$\text{LLM}_{\Theta}\left(y_i\right)^{\text{Q}} \xrightarrow{\text{finish}} y_{i+1} = \text{fetch}\left(z_i^N\right)^{\text{C}} \xrightarrow{\text{finish}} \text{LLM}_{\Theta}\left(y_{i+1}\right)^{\text{C}} \qquad (11)$$

These dependencies (10)(11) imposes a sequential workflow that increases decoding costs (Equation (3)). To implement a asynchronous pipeline, we argue that the operation issue$(f_{\theta_j})^{\text{C}}$ does not inherently depend on prior computations, as input buffers only require GPU buffer handlers instead of the actual value. Output buffers can also reuse handlers from previous predictions, with dependency (4) ensuring the prior computation is complete. Additionally, creating shaders and uniform buffers is unrestricted since their values are known, making preparation operation issue$(f_{\theta_j})^{\text{C, D}}$ independent of previous computations.

Based on our insights, we parallelize content timeline, device timeline and queue timeline by decoupling the preparation and the execution. This asynchronous pipeline allows the CPU to continuously prepare resources (Figure 2 ①) while the GPU keeps executing (Figure 2 ③), eliminating unnecessary blocking. This pipeline could be further improved by delaying fetching stages (Figure 2 ④) to reduce the cost of fetching results. To appropriately feedback predictions in time, we issue fetch requests every time a threshold $I$ of token are predicted. Fetching dependency of our pipeline is converted from (7) to:

$$y_{nI} = \text{fetch}\left(z^{N}_{(n-1)I,(n-1)I+1,\dots,nI}\right)^{C} \xrightarrow{\text{finish}} \hat{r}_j = \text{issue}\left(f_{\theta_j}\right)^{C},$$
(12)

where $n = 1, 2, \dots$. The overall dependencies of our pipeline can be expressed as:

$$\text{LLM}_{\Theta}\left(y_i\right)^{C} \xrightarrow{\text{finish}} \text{LLM}_{\Theta}\left(y_{i+1}\right)^{C},$$
(13)

$$\text{LLM}_{\Theta}\left(y_i\right)^{Q} \xrightarrow{\text{finish}} \text{LLM}_{\Theta}\left(y_{i+1}\right)^{Q}.$$
(14)

Through applying this optimization, the time overhead of each decoding step decreases from (3) to:

$$T = \frac{1}{I}\left(\delta T^{C, D}_{\text{prep}} + I \cdot T^{Q}_{\text{exec}} + T^{C, D}_{\text{fetch}}\right).$$
(15)

## 5 Evaluation

We first describe the experimental setup (Section 5.1). Then we conduct a performance analysis across heterogeneous models and devices (Section 5.2), followed by the results of ablation study (Section 5.3) and hyper-parameter sensitivity study (Section 5.4).

### 5.1 Setup

**Implementation**. Since MediaPipe LLM is not fully open-source [2] and Transformer.js has not officially released WebGPU support, we modify WebLLM (version 0.2.46) to implement WeInfer.

For the WebGPU uniform buffer reuse optimization, we store uniform buffers in a map using their values as keys in the first decoding step, allowing reuse in subsequent steps when the values remain unchanged. WebLLM suffers from synchronous blocking due to `await` statements when fetching results through WebGPU, forcing sequential processing. We modify these operations to handle them asynchronously and shift post-processing from the CPU to the GPU, enabling parallel preparation and execution. Additionally, we introduce a scheduler to manage preparation and fetching, allowing timely feedback while minimizing the overhead of fetching.

WeInfer seamlessly inherits the existing operator-level optimizations of WebLLM and extends the framework's capabilities. The modularity of our approach allows easy adaptation to other Web-based LLM inference frameworks facing similar issues.

**Baseline**. We compare WeInfer with WebLLM (version 0.2.46). To ensure fairness, we modify WebLLM to perform post-processing on the GPU and use greedy sampling with identical input prompts for both frameworks.

**Environment**. We measure the decoding speed of WeInfer and WebLLM in two scenarios: model heterogeneity (i.e., across different LLM architectures and parameter sizes) and device heterogeneity (i.e., across different GPU hardware).

To assess the impact of model heterogeneity, we evaluate Qwen2 [50], Llama3 [11], and SmolLM [2] with parameter sizes ranging from 135M to 8B on an RTX 3060 GPU running Windows OS.

To assess the impact of device heterogeneity, we evaluate performance on GPUs with different computational capabilities, including Intel UHD Graphics 630, Apple M2, GTX 1660 Ti, RTX 3060, and

---

[2]MediaPipe LLM leverages a GraphRunner to infer LLM, whose detailed implementation of the underlying WebAssembly module is hidden, posing difficulties to modify the inference procedure.

**Table 3: Average per-token decoding speed (ms/token) comparison across heterogeneous LLMs**

| Model | WebLLM | WeInfer | Boost |
|---|---|---|---|
| **SmolLM-135M-q4f16** | 27.72 | **10.17** | 2.73× |
| **SmolLM-135M-q4f32** | 27.72 | **11.68** | 2.37× |
| **Qwen2-0.5B-q4f16** | 22.49 | **9.67** | 2.33× |
| **Qwen2-0.5B-q0f32** | 21.28 | **12.10** | 1.76× |
| **TinyLlama-1.1B-q4f16** | 21.09 | **9.76** | 2.16× |
| **Qwen2-1.5B-q4f16** | 24.57 | **14.88** | 1.65× |
| **Qwen2-1.5B-q4f32** | 24.18 | **18.07** | 1.34× |
| **Qwen2-7B-q4f16** | 32.75 | **29.19** | 1.12× |
| **Llama3-8B-q4f16** | 71.51 | **64.35** | 1.11× |

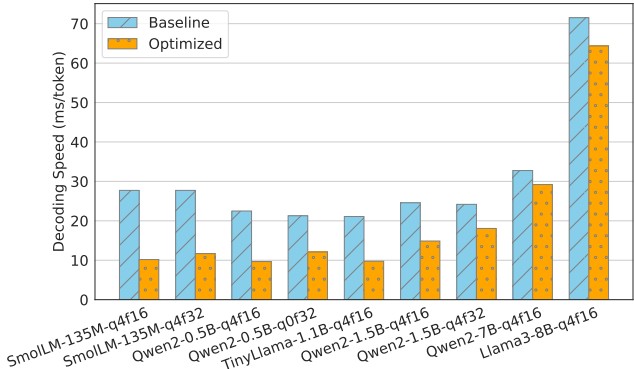

**Figure 4: Performance comparison of inferring heterogeneous models**

RTX 4090. To explore performance across different operating systems and GPU drivers, we run experiments on Linux (with RTX 4090), macOS (with Apple M2), and Windows (with other GPUs). Models are restricted due to the memory constraints of lower-end GPUs. We are unable to perform inference with half-precision float (f16) models on Linux, as the device does not support WebGPU's `shader-f16` feature. These results are indicated as N/A.

We use Google Chrome (version 131.0) as the browser, which currently offers the best WebGPU support. Firefox Nightly is excluded due to its resource limitations that are incompatible with the requirements of WebLLM (e.g., `maxComputeWorkgroupStorageSize`).

### 5.2 Result Analysis

*5.2.1 Impact of Model Heterogeneity.* Table 3 and Figure 4 present the performance comparisons across different LLMs. WeInfer demonstrates a significant speedup across all evaluated models, with boosts ranging from 1.11× to 2.73×.

For the smallest model, SmolLM-135M with half-precision float (f16), WeInfer achieves a decoding speed of 10.17 ms/token, while WebLLM slows to 27.72 ms/token, resulting in a 2.73× speedup. For larger models like Llama3-8B and Qwen2-7B, WeInfer achieves decoding speeds of 64.35 ms/token and 29.19 ms/token, respectively, while WebLLM performs decoding at speeds of 71.51 ms/token and 32.75 ms/token. The boost is less pronounced for larger models

**Table 4: Average per-token decoding speed (ms/token) comparison across heterogeneous devices**

| GPU Hardware | SmolLM-135M-q4f16 | | | SmolLM-135M-q4f32 | | | Qwen2-0.5B-q0f32 | | | Qwen2-1.5B-q4f32 | | |
|---|---|---|---|---|---|---|---|---|---|---|---|---|
| | WebLLM | WeInfer | Boost | WebLLM | WeInfer | Boost | WebLLM | WeInfer | Boost | WebLLM | WeInfer | Boost |
| **RTX 4090** | N/A | N/A | N/A | 28.27 | **8.89** | 3.18× | 26.85 | **8.12** | 3.31× | 31.72 | **8.43** | 3.76× |
| **RTX 3060** | 27.72 | **10.17** | 2.73× | 27.72 | **11.68** | 2.37× | 22.49 | **9.67** | 2.33× | 24.18 | **18.07** | 1.34× |
| **GTX 1660 Ti** | 54.02 | **31.98** | 1.69× | 56.07 | **32.08** | 1.75× | 47.05 | **23.68** | 2.01× | 63.29 | **35.14** | 1.80× |
| **Apple M2** | 17.51 | **8.75** | 2.01× | 17.32 | **8.93** | 1.94× | 32.79 | **29.38** | 1.12× | 33.05 | **29.21** | 1.13× |
| **Graphics 630** | 60.94 | **42.92** | 1.42× | 75.82 | **62.45** | 1.21× | 252.95 | **238.03** | 1.06× | 403.99 | **394.84** | 1.02× |

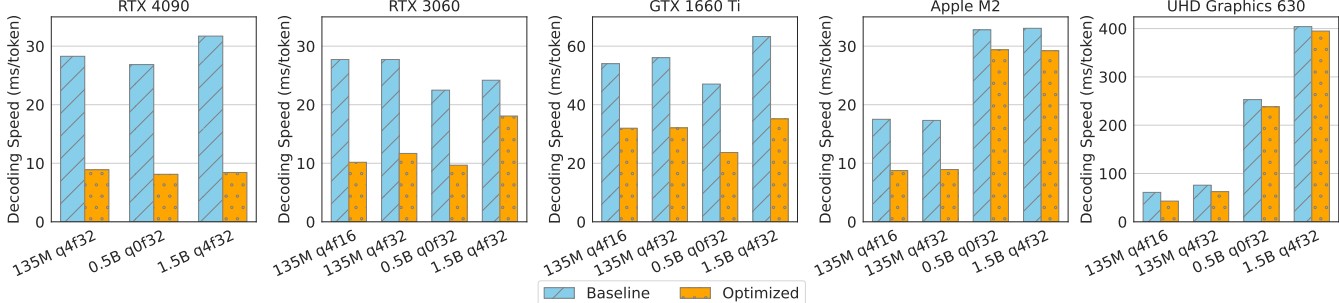

**Figure 5: Performance comparison of inference on heterogeneous devices**

due to the increasing computational demands, which shifts the bottleneck to GPU execution time $T_{\text{Exec}}^{\text{Q}}$. WeInfer primarily reduces latency in preparation and fetching, benefiting less from scenarios where GPU execution dominates the decoding speed.

### 5.2.2 Impact of Device Heterogeneity.
Table 4 and Figure 5 summarize the performance across different GPU hardware. The results indicate that WeInfer consistently improves decoding speed across all devices, with the most significant speedup observed on high-end GPUs, reaching a 3.76× boost.

On the RTX 4090, WeInfer achieves a decoding speed of 8.43 ms/token for Qwen2-1.5B, compared to WebLLM's 31.72 ms/token, resulting in a 3.76× speedup. On mid-range devices like the RTX 3060, WeInfer achieves a 1.34× improvement, reducing the decoding speed of Qwen2-1.5B from 24.18 ms/token (WebLLM) to 18.07 ms/token. On low-end GPUs like Intel UHD Graphics 630, the speedup is less pronounced (up to 1.21×), as GPU execution time dominates the overall decoding time. Additionally, we observe performance instability on the Apple M2 chip: while WeInfer achieves a 2.01× boost for SmolLM-135M, the boost drops to 1.12× for Qwen2-0.5B. Traces recorded by Chrome DevTool reveal that WeInfer suffers from increased execution time when inferring Qwen2, counteracting gains from reduced WebGPU-specific overhead. This may be due to the unique characteristics of Metal APIs [4] or Apple's GPU hardware.

### 5.3 Ablation Study
We perform ablation experiments on the RTX 3060 to evaluate the contribution of each optimization. We evaluate decoding speeds of various models under three conditions: using only uniform buffer reuse, using only asynchronous pipeline, and using both optimizations. The results are summarized in Table 5.

The results demonstrate that both optimizations contribute significantly to the improvement of decoding speed. For instance, with the Qwen2-1.5B model, uniform buffer reuse reduces decoding time by approximately 8 ms, while the asynchronous pipeline reduces it by 4 ms on average. The smaller SmolLM model sees a 12 ms improvement from buffer reuse and 4 ms from the asynchronous pipeline. Uniform buffer reuse exhibits more variability depending on model size and architecture, as its effectiveness is linked to the preparation time $T_{\text{prep}}^{\text{C,D}}$. In contrast, pipeline parallelization consistently reduces the synchronous blocking time $T_{\text{fetch}}^{\text{C,D}}$, which is less dependent on model size.

### 5.4 Hyper-parameter Sensitivity Study
WeInfer introduces a hyper-parameter $I$ in the asynchronous pipeline, controlling the number of tokens generated before the CPU fetches predictions from the result buffer. This fetch interval enables WeInfer to balance real-time feedback with task batching, enabling the GPU to perform multiple operations before synchronously waiting for results.

We conduct experiments on the RTX 3060 to evaluate the effect of different fetch intervals $I$ on decoding speed. The results are presented in Table 6.

The results show that performance fluctuates slightly with different fetch intervals, but the results improve across all intervals, demonstrating that our optimizations deliver significant improvements even when prioritizing real-time feedback (e.g., with $I = 4$). We identify that for smaller models like SmolLM-135M and Qwen2-0.5B, optimal performance is generally observed with $I \geq 16$, where

**Table 5: Ablation experiments on average per-token decoding speed (ms/token) using different optimizations**

| Model | WebLLM | Reuse | Pipeline | WEINFER |
|---|---|---|---|---|
| SmolLM-135M-q4f16 | 27.72 | 15.46 | 23.31 | **10.17** |
| SmolLM-135M-q4f32 | 27.72 | 15.42 | 23.42 | **11.68** |
| Qwen2-0.5B-q4f16 | 22.49 | 15.02 | 19.97 | **9.67** |
| Qwen2-0.5B-q0f32 | 21.28 | 14.58 | 18.82 | **12.10** |
| Qwen2-1.5B-q4f16 | 24.57 | 16.61 | 19.92 | **14.88** |
| Qwen2-1.5B-q4f32 | 24.18 | 19.80 | 22.52 | **18.07** |
| Qwen2-7B-q4f16 | 32.75 | 31.60 | 31.43 | **29.19** |
| Llama3-8B-q4f16 | 71.57 | 67.69 | 66.40 | **64.35** |

**Table 6: Sensitive analysis of the impact of fetch interval on average per-token decoding speed (ms/token)**

| Model | $I = 2$ | $I = 4$ | $I = 8$ | $I = 16$ | $I = 32$ |
|---|---|---|---|---|---|
| SmolLM-135M-q4f16 | 12.48 | 13.11 | 11.56 | **10.17** | 12.30 |
| SmolLM-135M-q4f32 | 13.30 | 13.02 | **11.68** | 11.70 | 11.70 |
| Qwen2-0.5B-q4f16 | 12.71 | 14.07 | 10.73 | 10.35 | **9.67** |
| Qwen2-0.5B-q0f32 | 13.25 | **12.10** | 12.85 | 12.35 | 12.62 |
| Qwen2-1.5B-q4f16 | 16.31 | **14.88** | 15.96 | 15.97 | 16.30 |
| Qwen2-1.5B-q4f32 | 19.50 | **18.07** | 19.45 | 19.85 | 20.47 |
| Qwen2-7B-q4f16 | 30.94 | **29.19** | 30.28 | 30.80 | 30.96 |
| Llama3-8B-q4f16 | 66.30 | **64.35** | 64.99 | 65.61 | 65.48 |

fetching overhead is sufficiently amortized. However, for larger models like Llama3-8B, increasing $I$ offers diminishing returns as execution time becomes the primary bottleneck.

## 6 Related Work

This section surveys existing work on LLM inference and optimizations for both browser and native environments, highlighting the limitations of existing methods in accelerating Web-based LLM inference.

### 6.1 Web-based LLM Inference Frameworks and Optimizations

Early efforts to run deep neural network inference in browsers [28], such as TensorFlow.js [14], WebDNN.js [23] and ONNX Runtime Web [29], laid the foundation for browser-based model execution. However, as LLMs present greater computational demands, specialized frameworks have emerged. Transformer.js [12] offers a high-level LLM inference API fully within browsers, similar to its Python counterpart. WebLLM [30], the first to support billion-parameter LLMs in browsers, and MediaPipe LLM [15], which was introduced by Google in 2024, both use WebGPU to enable GPU acceleration. Transformer.js is expected to adopt WebGPU in its upcoming version.

Efforts to accelerate in-browser inference tackle challenges related to heterogeneous devices and constrained environments. Compilation frameworks like TVM, Ansor, and FlexTensor [8, 56, 57] employ automatic operator tuning for hardware-specific optimizations,

while NNJIT [25] enables just-in-time tuning for real-time improvements. Dynamic adaptation frameworks such as DeepAdapter [24] adjust inference processes based on device capabilities and network conditions. Additional approaches like PipeEngine [41] improve inference by partitioning models for parallel execution across Web Workers and WebGL, while WPIA [40] reduces initialization delays by distributing precompiled WebGL programs.

Many of these optimizations are integrated into current Web-based LLM inference frameworks. WebLLM leverages TVM for operator tuning, Transformer.js benefits from ONNX Runtime Web's subgraph fusion. MediaPipe LLM incorporates advanced GPU acceleration techniques like SIMD-level parallelism and optimized matrix multiplication via cooperative thread groups [17].

In contrast, WEINFER introduces a WebGPU-centric approach specifically designed to address the challenges posed by WebGPU and LLMs, differentiating itself from optimizations that primarily focus on operators or model architecture.

### 6.2 On-device LLM Inference Optimizations

Edge devices face memory and computational constraints that limit LLM inference. Techniques such as quantization and knowledge distillation [21, 33, 51] reduce model size and complexity, while frameworks like FlexGen and LLM-in-a-Flash [1, 38] tackle parameter offloading when memory is insufficient. Further advancements [39, 53] optimize offloading while integrating techniques such as mixture of experts and speculative execution.

For acceleration, distributed computing approaches, such as LinguaLinked, Galaxy, and PipeLLM [27, 52, 55], leverage nearby trusted computing resources to achieve collaborative LLM inference. PowerInfer-2 [49] optimizes matrix operations for mobile devices, while mllm-NPU [48] and NeuPIMs [22] explore NPUs to accelerate LLM.

These methods focus on refining operators or inference algorithms for specific edge devices, or leveraging distributed resources. WEINFER complements them by concentrating on in-browser inference, offering a novel solution that handles the unique challenges of Web environments.

## 7 Conclusion

In this paper, we analyzed the GPU efficiency of existing Web-based LLM inference frameworks and identified the underlying performance bottlenecks introduced by the suboptimal utilization of WebGPU. We proposed WEINFER, a novel framework to address these limitations. Compared to previous efforts that primarily focused on optimizing neural network operators or model architectures, our approach targets reducing the overhead associated with the process isolation mechanism of browsers by leveraging the advanced features of WebGPU.

The experimental results demonstrated significant and consistent improvements in decoding speed across a wide range of models and devices, confirming the effectiveness of WEINFER in diverse scenarios. Ablation studies and hyper-parameter analysis demonstrated the effectiveness of each optimization and the robustness of WEINFER. Moving forward, we aim to develop adaptive acceleration strategies tailored to specific hardware configurations to better harness the computational capabilities of edge devices.

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

## A  BROWSER CONSTRAINTS ON LEVERAGING GPU

Modern browsers utilize a process-based isolation architecture, segregating kernel processes from content processes where Web applications run. This architecture ensures a clear separation between the Web domain and the local environment, protecting against security threats by confining potentially malicious applications within the browser sandbox. Additionally, this model ensures that failures in individual applications do not compromise the entire browser. This architecture has been adopted by most browsers such as Chrome, Internet Explorer, Firefox, and Microsoft Edge.

In browsers with such architecture, multiple content processes are created for individual Web applications. These untrusted processes are restricted in their access to system resources, including the device's GPU. Any GPU access requested by a Web application must be mediated via inter-process communication (IPC) with a kernel-level GPU process. This GPU process operates under reduced sandboxing constraints compared to content processes and is typically shared by multiple applications. To maintain security and integrity, the GPU process validates every message it receives, preventing unauthorized access to GPU memory or exploitation of the shared GPU process. Additionally, the GPU driver is loaded directly into the GPU process, as the GPU process is responsible for all direct communication with the GPU hardware. This means that GPU buffers are mapped only to the virtual memory space of the GPU process, making them inaccessible to content processes and limiting the efficiency of buffer reads.

As a result, the design of WebGPU focus on ensuring security and maintaining sandbox abstraction to be implementable and efficient in modern browsers with this single GPU process architecture.

- WebGPU enforces strict validation within the GPU process when preparing computational resources. The GPU process must validate all messages according to WebGPU standards to restrain compromised content processes from malicious memory access or commands execution.
- Unlike native GPU APIs that are able to map GPU buffers directly into the memory space of applications, WebGPU requires an additional staging step to read GPU buffers as only the GPU process holds GPU drivers. Specifically, when a Web application issues a map request, shared memory is required to be allocated within the GPU process. This shared memory acts as both the destination of mapping from the GPU memory and the source of mapping to the memory of application, facilitating the transfer of GPU buffer data to a space accessible by the content process. This 2-stage approach ensures safely transferring ownership of the buffer from the GPU to the CPU while increasing overheads on fetching results from the GPU.

