# OpenReview forum: "WeInfer: Unleashing the Power of WebGPU on LLM Inference in Web Browsers"
_ACM.org/TheWebConf/2025/Conference — WWW 2025 Poster_

### Official Review · Reviewer_64jk · 2024-11-14

**Novelty:** 5
**Technical Quality:** 6

**Review:**

This paper proposes WeInfer, a framework for LLM inference in web browsers. It observes that existing web inference frameworks are inefficient because they have high overheads for resource preparation and result fetching. WeInfer reduces the two overheads by reusing the uniform buffers and pipelining preparation and resource utilization. Experiment results show that WeInfer accelerates the decoding speed compared with WebLLM.

Strength

S1: LLM inference in web browsers is an important application scenario.

S2: The inefficiencies of existing frameworks are thoroughly analyzed with profiling results.

S3: The technical solutions, although not surprising, make sense.

S4: The empirical performance is good.


Weakness

W1: Experiment settings can be introduced more clearly. For Table 3, which GPU is used? Can the 2 largest models be used to enhance Table 4?

W2: In Table 6, the inference time fluctuates when increasing I. Intuitively, the inference time should decrease with I. The reason should be discussed.

W3: Please compare the GPU utilization of WeInfer and WebLLM.

**Questions:**

Do the optimizations of WeInfer introduce any overheads or negative effects?

**Reviewer Confidence:**

3: The reviewer is confident but not certain that the evaluation is correct

**Scope:**

4: The work is relevant to the Web and to the track, and is of broad interest to the community

---

### Official Review · Reviewer_4uZB · 2024-11-26

**Novelty:** 3
**Technical Quality:** 5

**Review:**

## Summary
This work introduces **WeInfer**, a novel framework that optimizes large language model (LLM) inference in web browsers by leveraging WebGPU's capabilities. It tackles inefficiencies in existing frameworks through two main contributions: buffer reuse strategies and an asynchronous pipeline design. The authors conduct extensive evaluations across multiple models and hardware, demonstrating substantial speedups over state-of-the-art systems like WebLLM.

## Strengths
1. **Targeted Optimization**: The paper identifies and addresses inefficiencies in WebGPU utilization with two clear strategies: buffer reuse and asynchronous pipelines.
2. **Comprehensive Evaluation Scope**: The work tests on multiple models and devices, showcasing general applicability.
3. **Relevance**: It aligns with growing interest in browser-based LLM inference for improved privacy and reduced latency.

## Weaknesses
1. **Limited Novelty**: The work lacks sufficient ablation studies to isolate the contributions of its two core innovations. The improvements seem to primarily stem from buffer reuse, making it challenging to assess the overall novelty and value.
2. **Inadequate SOTA Comparison**: The experiments are primarily compared against WebLLM, without rigorous evaluation against other frameworks like MediaPipe LLM, limiting confidence in the claimed advancements.
3. **Incomplete Discussion of Practical Challenges**: The paper does not deeply discuss potential challenges, such as compatibility issues or memory constraints, that might arise when deploying WeInfer on resource-limited devices.

**Questions:**

### Questions for the Authors

1. **Necessity of LLM**: Why is the inclusion of LLM essential to your solution? If large models are abstracted, would your framework still provide meaningful performance improvements? Could you elaborate on how important LLM is for WeInfer?

2. **Comparison with SOTA**: Why were frameworks like MediaPipe LLM not included in direct comparisons? Could you clarify how WeInfer performs against them in terms of decoding speed, memory usage, and other relevant metrics?

3. **Broader Metrics**: Beyond decoding speed, have you evaluated WeInfer's impact on memory consumption, energy efficiency, or scalability for larger models? If not, could you explain why and whether these factors could present limitations?

4. **Practical Deployment**: How does WeInfer handle compatibility issues with older GPUs or browsers with limited WebGPU support? Are there any known constraints that could affect its deployment in real-world scenarios?

5. **Asynchronous Pipeline Impact**: Could you elaborate on how the asynchronous pipeline contributes to the overall performance improvements? Specifically, in Table 5, buffer reuse already achieves significant reductions in preparation costs, while the pipeline optimization alone shows relatively modest gains. However, when combined, the performance improves substantially. Could you explain the synergy between these two techniques and why their integration leads to greater overall improvements?

6. **Real-World Applications**: Have you tested WeInfer in any end-user applications, such as browser-based chatbots or edge devices? If so, could you share insights or challenges observed in these settings?

**Reviewer Confidence:**

3: The reviewer is confident but not certain that the evaluation is correct

**Scope:**

4: The work is relevant to the Web and to the track, and is of broad interest to the community

---

### Official Review · Reviewer_jSz3 · 2024-11-27

**Novelty:** 6
**Technical Quality:** 5

**Review:**

Quality:

1. The paper proposes a framework that optimizes LLM inference performance in web browsers using WebGPU, and effectively improves inference speed through buffer caching and asynchronous pipeline methods.
2. The experiments are comprehensive, covering models of different scales (ranging from 135M to 8B parameters) and various hardware platforms (from low-end to high-end GPUs), validating the generalizability and effectiveness of the method.
3. The optimization methods appear to be general.

Clarity: The writing is very clear.

Originality and Importance:

1. Currently, there is limited research on inference with WebGPU. This paper provides an in-depth analysis of the inference bottlenecks on WebGPU and proposes corresponding optimization strategies.

**Questions:**

1. Could the innovation of the proposed optimization method under the specific constraints of WebGPU be made clearer?
2. Can you further explain the advantages of using WebGPU compared to directly using native GPU APIs?

**Reviewer Confidence:**

4: The reviewer is certain that the evaluation is correct and very familiar with the relevant literature

**Scope:**

4: The work is relevant to the Web and to the track, and is of broad interest to the community

---

### Official Review · Reviewer_feVq · 2024-12-05

**Novelty:** 5
**Technical Quality:** 5

**Review:**

This paper,WeInfer: Unleashing the Power of WebGPU on LLM Inference in Web Browsers,optimizes the WebGPU technique on LLM inference from main two direction: increasing caches for buffer reuse in preparation process and using asynchronous pipeline for parallelization of preparation and execution.

Quality: The new WeInfer is designed based on the exploration in existing WebLLM, finding some constrains that decrease inference speed. WeInfer use ideas like cache or asynchronous pipeline to improve inference performance, which is reasonable. The whole structure of this paper is clear.

Clarity: The design ideas are explained in a clear way with some mathematical expressions or codes provided. However, the experiment result tables may not clear only with numerical data. Some figures are too far away from the related context.

Originally: This paper targets at different area compared with previous studies. However, the general ideas are not novel enough as same ideas have discussed in other LLM inference scenarios.

Significance: The new framework WeInfer may provide potential improvement for small models in Web-based inference. However, for larger models this work can not help a lot.

Pros:
(1) Existing frameworks like WebLLM are explored and constrains are discovered in the process.
(2) Improve performance by merging old technique with current problems.
(3) Development in different area compared with previous study so that potentially easy for combination with other framework.

Cons:
(1) Improvements limit at small models.
(2) Other costs besides time are not discussed.

**Questions:**

(1) Can this work combine with other improvements and get a better performance for Web-based LLM inference?
(2) How is the robustness of this WebInfer? Will the new framework keep same inference quality with a shorter token generation delay?
(3) Can the framework be developed further? Because the performance in large model is not good and this may be a vital problem in future.

**Reviewer Confidence:**

3: The reviewer is confident but not certain that the evaluation is correct

**Scope:**

4: The work is relevant to the Web and to the track, and is of broad interest to the community